# Recent Trends and In-Hospital Mortality of Transarterial Chemoembolization (TACE) in Germany: A Systematic Analysis of Hospital Discharge Data between 2010 and 2019

**DOI:** 10.3390/cancers14092088

**Published:** 2022-04-22

**Authors:** Sarah Krieg, Tobias Essing, Andreas Krieg, Christoph Roderburg, Tom Luedde, Sven H. Loosen

**Affiliations:** 1Clinic for Gastroenterology, Hepatology and Infectious Diseases, University Hospital Düsseldorf, Medical Faculty of Heinrich Heine University Düsseldorf, 40225 Düsseldorf, Germany; sarah.krieg@med.uni-duesseldorf.de (S.K.); t.essing@web.de (T.E.); christoph.roderburg@med.uni-duesseldorf.de (C.R.); 2Paracelsus Medical University, Klinikum Nürnberg, 90419 Nürnberg, Germany; 3Department of Surgery (A), Heinrich-Heine-University and University Hospital Düsseldorf, 40225 Düsseldorf, Germany; andreas.krieg@med.uni-duesseldorf.de

**Keywords:** hepatocellular carcinoma, HCC, cholangiocarcinoma, CCA, liver metastases, biliary tract cancer, loco-ablative therapy, cancer

## Abstract

**Simple Summary:**

The minimally invasive procedure of transarterial chemoembolization (TACE) represents a standard oncologic procedure for the treatment of intermediate-stage hepatocellular carcinoma, cholangiocarcinoma, and liver metastases. Up to now, comprehensive data on recent trends and post-interventional in-hospital mortality have been largely lacking. Therefore, the aim of our study was to provide a systematic overview of the different indications and embolization techniques, as well as to identify factors associated with increased in-hospital mortality. For this purpose, we retrospectively examined 49,595 individual cases in Germany, based on standardized hospital discharge data from the German Federal Statistical Office, over the period 2010 to 2019. As a result of our study, we were able to show a correlation of in-hospital mortality with the presence of various complications, the type of embolization used, and the annual case volume. Consequently, our study may help to reduce the mortality of this therapeutic procedure in the future.

**Abstract:**

(1) Background: Transarterial chemoembolization (TACE) is a minimally invasive procedure, characterized by the selective occlusion of tumor-feeding hepatic arteries, via injection of an embolizing agent and an anticancer drug. It represents a standard of care for intermediate-stage hepatocellular carcinoma (HCC), and it is also increasingly performed in cholangiocarcinoma (CCA), as well as in liver metastases. Apart from the original method, based on intra-arterial infusion of a liquid drug followed by embolization, newer particle-based TACE procedures have been introduced recently. As yet, comprehensive data on current trends of TACE, as well as its in-hospital mortality in Germany, which could help to further improve outcome following TACE, are missing. (2) Methods: Based on standardized hospital discharge data, provided by the German Federal Statistical Office from 2010 to 2019, we aimed at systematically evaluating current clinical developments and in-hospital mortality related to TACE in Germany. (3) Results: A total of 49,595 individual cases undergoing TACE were identified within the observation period. The overall in-hospital mortality was 1.00% and significantly higher in females compared to males (1.12 vs. 0.93%; *p* < 0.001). We identified several post-interventional complications, such as liver failure (51.49%), sepsis (33.87%), renal failure (23.9%), and liver abscess (15.87%), which were associated with a significantly increased in-hospital mortality. Moreover, in-hospital mortality significantly differed between the underlying indications for TACE (HCC: 0.83%, liver metastases: 1.22%, and CCA: 1.40%), as well as between different embolization agents (liquid embolization: 0.80%, loaded microspheres: 0.92%, spherical particles: 1.54%, and non-spherical particles: 2.84%), for which we observed large geographic differences in their frequency of use. Finally, in-hospital mortality was significantly increased in centers with a low annual TACE case volume (<15 TACE/year: 2.08% vs. >275 TACE/year: 0.45%). (4) Conclusion: Our data provide a systematic overview of indications and embolization methods for TACE in Germany. We identified a variety of factors, such as post-interventional complications, the embolization method used, and the hospitals’ annual case volume, which are associated with an increased in-hospital mortality following TACE. These data might help to further reduce the mortality of this routinely performed local-ablative procedure in the future.

‡ These authors share senior authorship.

## 1. Introduction

Transarterial chemoembolization (TACE) is a minimally invasive radiologic procedure, first described by Yamada et al. in 1983 [1], which represents a standard of care for intermediate-stage hepatocellular carcinoma (HCC) and is also increasingly performed in patients with cholangiocarcinoma (CCA) or liver metastases, especially originating from colorectal carcinoma, melanoma, or neuroendocrine tumors [2]. TACE is based on the fact that more than 80% of the liver perfusion is supplied by the portal circulation, whereas liver tumors are mainly fed from the hepatic artery [3]. Therefore, it aims to selectively obstruct tumor-feeding hepatic arteries by injection of an embolizing agent, in combination with a chemotherapeutic substance, causing necrosis of tumor tissue via cytotoxic and ischemic effects, while preserving the healthy surrounding liver parenchyma [4,5]. Although treatment with TACE alone usually does not achieve a cure with the exception of small, solitary HCC lesions, prospective randomized trials have shown a significant survival benefit for TACE compared to best supportive care (BSC) [6,7]. Regarding the selection of embolization and chemotherapeutic agent, no generally accepted standards exist to date. Originally, Lipiodol^®^ (ethiodized oil) and Gelfoam^®^ (gelatin sponge) were used to perform conventional TACE (cTACE). Recently, a newer generation of TACE embolization agents have been developed consisting of polymers that could, themselves, be composed of microspheres, beads, or hydrogels [8,9,10]. Although TACE is known to be a safe procedure in general, it is not free of complications. In this context, among others, cholecystitis, acute pancreatitis, liver abscess, as well as liver and kidney failure have been reported [11,12,13]. A number of risk factors associated with serious post-interventional complications and a rise in mortality have been discussed in the literature, based mainly on laboratory, clinical, and radiological parameters [14]. However, comprehensive data on recent trends, as well as in-hospital mortality, following TACE in Germany, which could help to identify parameters associated with an increased mortality with the aim to improve patient outcomes, are largely missing.

We therefore analyzed standardized hospital discharge data, provided by the Federal Statistical Office of patients undergoing TACE between 2010 and 2019, to systematically assess recent clinical trends, post-interventional complications, and in-hospital mortality of TACE in Germany.

## 2. Materials and Methods

### 2.1. Study Design

We retrospectively assessed current trends, in-hospital mortality, and related parameters associated with TACE in Germany based on standardized hospital discharge data provided by the Federal Statistical Office (DESTATIS; Wiesbaden, Germany). For this purpose, a contract for remote data collection and analysis was concluded between the Federal Statistical Office and University Hospital Düsseldorf in 2020. The study was conducted in accordance with the guidelines of the Declaration of Helsinki. As the data were fully anonymized, no additional ethics approval was required.

### 2.2. Patient Selection Criteria and Variables

Patients were selected based on the International Statistical Classification of Diseases and Related Health Problems (ICD), a medical classification system of the World Health Organization (WHO), and on operation and procedure codes (OPS codes). Patients undergoing TACE were identified by ICD-10-codes C22.0 (HCC), C78.7 (liver metastases) or C21.0 (CCA) in main or secondary diagnosis and by the following OPS codes: (1) selective embolization with particles: OPS 8-836.ka: percutaneous transluminal vascular intervention + 8-83b.10: drug-eluting particles (e.g., DC Beads^®^), (2) selective embolization with particles: OPS 8-836.ka: percutaneous transluminal vascular intervention + 8-83b.12: Non-spherical particles (e.g., Contour™), (3) selective embolization with particles: OPS 8-836.ka: percutaneous transluminal vascular intervention + 8-83b.1: Other spherical particles (e.g., Embozene™), or (4) selective embolization with embolizing liquids: OPS 8-836.9a: percutaneous transluminal vascular intervention + 8-83b.2x: Other liquids. In total, 49,595 individual cases were identified an included into analysis. Note that, in some analyses, all three types of particle embolization were combined into a single category prior to being compared with liquid embolization. Clinical and demographic variables considered for analysis were as follows: age, sex, federal state, secondary diagnoses, i.e., liver failure (ICD-10-code: K72), acute pancreatitis (K85), cholecystitis (K81), liver abscess (K75), duodenitis (K29.8), gastritis (K29), acute renal failure (N17), sepsis (A02; A20; A26; A32; A40; A41; A42; B37), and liver cirrhosis (K70; K71; K74). Hospital length of stay and in-hospital mortality, defined as the proportion of patients discharged for “death”, were assessed as additional variables. In relation to the annual TACE case volume, treatment centers performing TACE were categorized into four groups, as follows, based on quartiles; (low case volume (LCV) hospitals: 1–29 cases/year, medium-low case volume (MLCV) hospitals: 30–85 cases/year, medium-to-high case volume (MHCV) hospitals: 86–174 cases/year, and high case volume (HCV) hospitals: >174 cases/year. Next, centers were separated into octiles by annual case volume. Detailed information is presented in Table 1.

### 2.3. Statistical Analysis

We performed all statistical analyses, via remote data access, at the Federal Statistical Office (DESTATIS; Wiesbaden, Germany) using the statistical program SPSS (IBM Corporation, Armonk, NY, USA) and the spreadsheet program Excel (Microsoft Corporation, Redmond, WA, USA). Cross tabulations were generated for the analysis of descriptive data. Pearson’s chi-square test was applied for comparison of binary variables (e.g., death in hospital yes/no). Analyses of changes in the dependent and independent variables over time were performed using Pearson’s R and linear regression analysis. To compare for differences of metric variables between patient groups, we used the Kruskal–Wallis Test. Exclusively, two-tailed statistical tests were used. It was considered that a *p*-value < 0.05 was statistically significant.

## 3. Results

### 3.1. Current Trends of TACE in Germany between 2010 and 2019

A total of 49,595 individual cases undergoing TACE were identified and included into analyses (Table 1). For the majority of patients, the underlying diagnosis for TACE was HCC (*n* = 33,726; 68.0%). Liver metastases represented the second most common etiology (*n* = 13,578; 27.4%), while cholangiocarcinoma (CCA) made up the smallest percentage (*n* = 2291; 4.6%, Figure 1A). During the observation period, most interventions were recorded in 2015. Individual cases increased steadily from 2010 to 2015 and decreased in the following years, without significant alterations within the spectrum of etiologies (Figure 1B). Next, we evaluated the number of performed TACE procedures according to different embolization agents. The most frequently performed TACE procedure was selective embolization with liquids (*n* = 24,839; 50.08%), followed by selective embolization with drug-eluting particles (e.g., DC-Beads^®^, HepaSphere™; *n* = 15,670; 31.6%). Other procedures using non-spherical particles (e.g., Contour SE Particles™; *n* = 7926; 15.98%) and other spherical particles (e.g., Embozene™; *n* = 1160, 2.34%) were performed less often (Figure 1C and Appendix A). Focusing on the geographical distribution in Germany, most TACE cases were registered in North Rhine-Westphalia (*n* = 9162), followed by Baden-Württemberg (*n* = 6777) and Bavaria (*n* = 5776). The lowest number of TACE procedures was recorded in Brandenburg (*n* = 52) and Bremen (*n* = 87, Table 1 and Figure 1D). Comparing the ratio of particle embolization (including loaded microspheres, non-spherical particles, and other spherical particles) to the total number of TACE in each state, it was highest in Thuringia (89.0%), followed by North Rhine-Westphalia (73.0%), Mecklenburg-Western Pomerania (69.2%), and Brandenburg (65.4%), compared to a lower proportion of particle embolization in Saarland (7.7%), Berlin (24.0%), Schleswig-Holstein (29.2%), and Hesse (29.3%) (Figure 1D). Regarding sex distribution, the majority of patients were male (73.7%), and the sex ratio did not change during the observation period (Table 1 and Appendix A).

### 3.2. In-Hospital Mortality Rates following TACE in Germany

The overall in-hospital mortality rate, following TACE within the observation period, was 1.00% (Figure 2A, Table 1). There was no significant change of in-hospital mortality during the observation period (Figure 2B), although mortality rates numerically increased from 0.82% in 2011 to 1.19% in 2019 (Figure 2B). In-hospital mortality was significantly higher in female patients compared to males (1.12 vs. 0.93%; *p* < 0.001, Figure 2C). We did not observe a significant correlation between in-hospital mortality and age, although mortality numerically increased with higher age (Figure 2D). Moreover, our data showed a correlation between in-hospital mortality and the underlying tumor entity. As such, in-hospital mortality was lowest in HCC patients (0.89%) and significantly higher in patients with liver metastases (1.22%) or CCA (1.40%, Figure 2E). When comparing the different embolization agents, in-hospital mortality was lowest for liquid embolizations and loaded microspheres (0.80 and 0.92%), and it was significantly higher for spherical particles (1.54%) and non-spherical particles (2.84%, Figure 2F). Finally, in-hospital mortality did not significantly differ between the federal states in Germany (Appendix A).

### 3.3. Post-Interventional Complications following TACE and Their Impact on In-Hospital Mortality

We next evaluated common post-interventional complications following TACE in terms of different embolization agents. In general, most complications occurred with a low frequency of less than one percent (Figure 3A). Interestingly, the majority of complications including liver abscess (*p* < 0.001), cholecystitis (*p* = 0.005), pancreatitis (*p* < 0.001), liver failure (*p* = 0.001), renal failure (*p* = 0.018), and sepsis (*p* = 0.019) occurred significantly more often in patients receiving particle embolization (including loaded microspheres, non-spherical particles, and other spherical particles) compared to liquid embolization (Figure 3A). Subsequently, we evaluated whether post-interventional complications had an influence on in-hospital mortality. Here, we observed that the majority of post-interventional complications significantly increased in-hospital mortality, with liver failure (51.49 vs. 0.76%), sepsis (33.87 vs. 0.79%), acute renal failure (23.19 vs. 0.75%), and liver abscess (15.87 vs. 0.97%) being the complications associated with the largest increase of in-hospital mortality (Figure 3B). Of note, in-hospital mortality was comparable among patients with or without pre-existing liver cirrhosis (Figure 3C). We finally assessed the duration of hospital stays following TACE, according to the different embolization agents. Here, the average duration of hospitalization ranged from 4.18 to 6.40 days, with a significantly shorter duration of hospitalization for embolization with liquids compared to embolization with particles (Figure 3D).

### 3.4. Influence of Hospital Case Volume on In-Hospital Mortality

Finally, we evaluated a potential association between the hospitals’ annual TACE case volume and in-hospital mortality. Therefore, we divided treatment centers into four groups based on the quartiles of annual case volume (low case volume (LCV) hospitals: 1–29 cases/year, medium-low case volume (MLCV) hospitals: 30–85 cases/year, medium-high case volume (MHCV) hospitals: 86–174 cases/year, and high case volume (HCV) hospitals: >98 cases/year (Table 1). Importantly, we observed that in-hospital mortality, stepwise and significantly, decreased from 1.6% in LCV hospitals to only 0.65% in HCV hospitals (Figure 4A). Of note, MHCV and HCV hospitals had a comparable in-hospital mortality (Figure 4A). To further elaborate on possible subtle differences in case volume and in-hospital mortality, hospitals were next categorized into octiles according to their annual case volume (Table 1). Again, there was a significant decrease in in-hospital mortality from 2.1% (1–14 TACE/year) to 0.45% (>275 TACE/year, Figure 4B). Interestingly, the data show that especially hospitals with a very low annual TACE frequency of less than 15 are associated with a particularly high in-hospital mortality, while a further reduction in in-hospital mortality below 1% was observed from an annual case volume of 86 TACE procedures.

## 4. Discussion

TACE is based on the selective obstruction of tumor-supplying arteries inducing ischemic necrosis of the tumor, which, in turn, leads to damage of membrane receptors and increases the uptake of the chemotherapeutic agent [13]. It is indicated in hypervascular tumors and mostly performed in HCC, followed by liver metastases, while less commonly in CCA [2]. In this study, we showed an all-cause in-hospital mortality, following TACE, of 1%. Based on a systematic analysis of available data, we could identify parameters related to a higher in-hospital mortality. In this context, the presence of organ complications was considered to be the most significant risk factor. Although complications of TACE are rare, the most serious is liver failure, for which our analysis revealed an increase in in-hospital mortality to over 50%. Previous studies have already reported an association between in-hospital mortality and organ complications [2,13,15]. Consequently, preventing such events by early assessment and consistent treatment of potential organ complications should be an essential part of post-interventional patient care. A number of risk factors have been previously discussed in the literature as being associated with the development of serious post-interventional complications. Accordingly, relative contraindications to TACE have relied on the exclusion of patients with hepatic synthetic dysfunction and renal insufficiency, mainly based on laboratory criteria of one or more of the following: an elevation of serum bilirubin, aspartate aminotransferase, alanine aminotransferase, prothrombin time/international normalized ratio (INR), or creatinine level, as well as a decrease in platelet count or albumin [14]. Garwood et al. identified three significant predictors of TACE-related death, respectively, those being a rising INR, a MELD score ≥20, and a serum albumin level <2.0 g/L). Matching our data, which showed no difference in in-hospital mortality in patients with or without preexisting cirrhosis, Garwood et al., as well, have found cirrhosis not to be an independent predictor of death following TACE [14]. Moreover, our data revealed an association between in-hospital mortality and the underlying tumor diagnosis. In-hospital mortality was significantly lower when performed for HCC but higher when applied for the treatment of liver metastases or CCA. These findings might be due to the fact that patients with CCA and liver metastases already have a more advanced tumor stage, as well as a reduced general health status, when TACE is performed compared to HCC patients. Besides, our data demonstrated female sex as another risk factor, suggesting a sex-specific mortality risk. In contrast, our data did not show a significant association of in-hospital mortality and age, indicating that TACE is a safe treatment option even for older patients. These findings are consistent with data from previous studies that have evaluated TACE as a safe procedure in elderly patients without finding shortened survival or increased incidence of complications in those older than 75 years [16,17].

Regarding the selection of embolization and chemotherapeutic agent, no generally accepted standards exist to date. This might explain the fact that we observed a high variability concerning the most commonly used embolization agents even between the different federal states of Germany. Of note, the oldest and probably most performed liquid embolic agent is Lipiodol, an oily contrast agent that is known to simultaneously cause occlusion in blood vessels [2]. Alternatively, or in addition, solid embolization agents, e.g., microparticles, have been used more frequently in the last decade and are available in different diameters. In terms of production technology, older embolization agents, such as polyvinyl alcohol (PVA) particles or gelfoam, are presenting a wider spread of particle size, which might inhibit selective embolization. Recently, drug-eluting beads (DEB-TACE) have been developed for TACE in an effort to reduce systemic toxic side effects of chemotherapy [18]. For this, microspheres are loaded with a various number of drugs in order to release high concentrations, in a controlled and sustained manner, into the tumor tissue [19,20]. Finally, our data were able to show that, when comparing the different embolization agents, in-hospital mortality was lowest for liquid embolization and loaded microspheres (0.80 and 0.92%, respectively), while it was significantly higher for spherical particles (1.54%) and non-spherical particles (2.84%, Figure 2F). It is noteworthy that, in the comparison between embolization with particles of any type and embolization with liquids, a significantly higher rate on hospital complications was found when particle embolization was performed for TACE. However, a number of clinical trials in recent years have revealed the benefits of DEB-TACE over cTACE. Results from the PRECISION V trial, a multicenter randomized study comparing short-term outcomes of DEB-TACE and cTACE, have shown that DEB-TACE was associated with better tolerability, with significant reductions in liver toxicity, and reductions in chemotherapy-related adverse events [21]. In spite of the theoretical advantages of DEB-TACE, several clinical trials are still controversial as to whether DEB-TACE is superior to cTACE in terms of efficacy. A series of studies have already investigated the safety and efficacy of DEB-TACE for unresectable HCC [4,22,23,24,25,26], noting that few prospective and retrospective trials comparing cTACE and DEB-TACE were available [21,27,28]. Contemporary clinical trials and meta-analyses comparing these two treatment modalities have provided inconsistent conclusions, as some found no significant differences in tumor response rates in patients with unresectable HCC [29,30,31,32], while others demonstrated higher overall survival and better tumor response in patients with HCC treated with DEB-TACE compared with those receiving cTACE [33,34]. Recently, a meta-analysis of six RCTs concluded that the safety and efficacy profile of DEB-TACE was comparable to that of cTACE with no significant differences found between those two treatments, in terms of overall survival or serious complications [35]. Similarly, another meta-analysis published in 2021, which evaluated the safety, efficacy, and survival benefit of DEB-TACE versus cTACE in the treatment of patients with unresectable HCC, concluded that DEB-TACE was associated with better objective response and disease control, although pooled analysis revealed no significant benefit of DEB-TACE regarding complete or partial response, disease stability, and progression control (Alquahntani 2021). Indeed, DEB-TACE was presumed to have fewer severe complications, as well as lower 30-day and all-cause mortality [21,26]. Among their advantages, liquid embolization agents feature the ability to penetrate deeply into small-diameter target vessels, adapting to the lumen of any vessel size before transforming into a solid material. Therefore, in contrast to particle systems with narrow diameter ranges, liquid embolization agents are capable of occluding vessels of different diameters [36]. Accordingly, important criteria, such as improved biocompatibility, biodegradability, radiopacity, intrinsic bioactivity, and multidrug loading, require considerations in the development of multifunctional embolization agents. After all, our study was the first to investigate a correlation between the annual case volume of hospitals in Germany performing TACE and in-hospital mortality. Remarkably, there was a significant decrease in in-hospital mortality, from 2.1 to 1.1%, above an annual case volume of 15, with further reductions achieved for annual case volumes of 86 and above. When the number of cases exceeded 245, in-hospital mortality was even reduced to 0.45%. This finding highlights the importance that such complex procedures as TACE should be performed only in centers that have a minimum number of cases, in order to reduce in-hospital mortality and improve patient outcomes.

We acknowledge some limitations of our study. First, our retrospective database evaluation was able to provide information on in-hospital mortality for TACE only, but it was unable to draw any conclusion regarding later time points such as overall survival. This limitation is due to the anonymized data obtained from the Federal Statistical Office (DESTATIS; Wiesbaden, Germany), which does not allow individual follow up of patients after hospital discharge. Another limitation is that the analysis of in-hospital mortality rates is limited to their binary character by which no detailed assessment of the underlying cause or circumstances of death was feasible. Furthermore, no systematic analysis of coding quality in Germany has been performed to date, making a general conclusion as to how far coding, as a predictor for the further use of data in medicine and health care policy, reflects reality not entirely clear. Nevertheless, it can be considered that endpoints such as liver failure and death are little or not influenced by coding errors and correctly represent medical practice. Finally, our data are based on individual cases/TACE procedures and not individual patients. Therefore, no information was available on the number of patients who underwent TACE more than once, nor did our data include other factors of TACE, such as particle size, chemotherapy concentration, and treatment selectivity. Accordingly, to illuminate these factors in more detail, further prospective randomized controlled trials are needed. These future studies should also include multivariate analyses to further dissect the influence of individual factors on in-hospital mortality.

## 5. Conclusions

Together, our data provide, for the first time, a systematic overview of the indications, as well as the embolization methods, used for TACE in Germany within the last decade. We identified a variety of factors, such as the occurrence of different complications, the type of embolization used, or the annual case volume, which are associated with an increased in-hospital mortality of TACE and may help to further reduce the post-interventional mortality of this routinely performed procedure in the future.

## Figures and Tables

**Figure 1 cancers-14-02088-f001:**
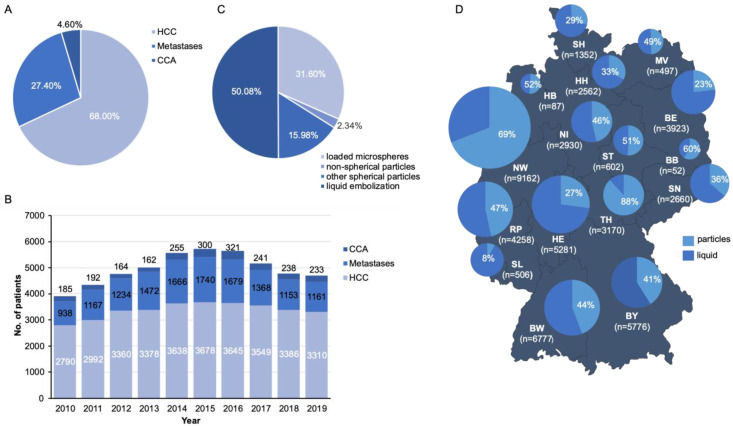
Current trends of TACE in Germany. (**A**) Percentage of underlying diagnosis for TACE (HCC: Hepatocellular Carcinoma, CCA: Cholangiocellular Carcinoma). (**B**) Total distribution of underlying diagnosis for TACE between 2010 and 2019. (**C**) Total distribution of TACE in terms of different embolization agent. (**D**) Prevalence of TACE between 2010 and 2019, including the proportion of particle embolization by percentage (BB: Brandenburg, BE: Berlin, BW: Baden-Württemberg, BY: Bavaria, HE: Hesse, HB: Bremen, HH: Hamburg, MV: Mecklenburg-Western Pomerania, NI: Lower Saxony, NW: North Rhine-Westphalia, RP: Rhineland-Palatinate, SH: Schleswig-Holstein, SL: Saarland, SN: Saxony, ST: Saxony-Anhalt, TH: Thuringia).

**Figure 2 cancers-14-02088-f002:**
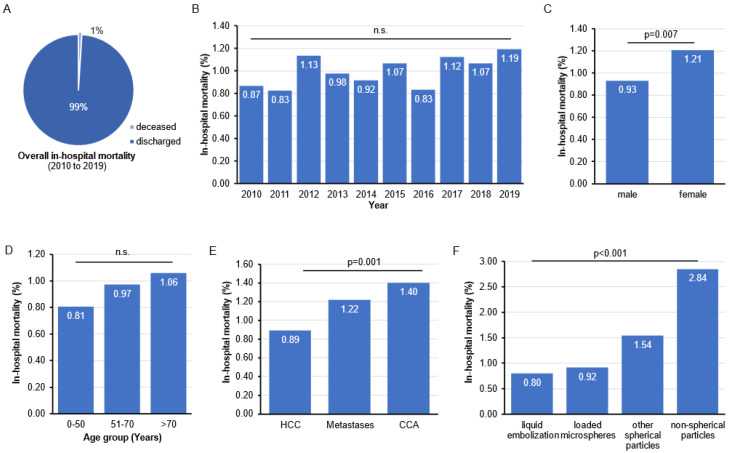
In-hospital mortality following TACE in Germany. (**A**) The overall in-hospital mortality between 2010 and 2019 was 1.00%. (**B**) No significant trend in in-hospital mortality was observed between 2010 and 2019. (**C**) In-hospital mortality rates are significantly lower in males compared to female patients. (**D**) There is no significant association of in-hospital mortality with age. (**E**) In-hospital mortality differs along the underlying tumor diagnosis (HCC: Hepatocellular Carcinoma, CCA: Cholangiocellular Carcinoma). (**F**) A significant difference in in-hospital mortality was observed between the different embolization agents.

**Figure 3 cancers-14-02088-f003:**
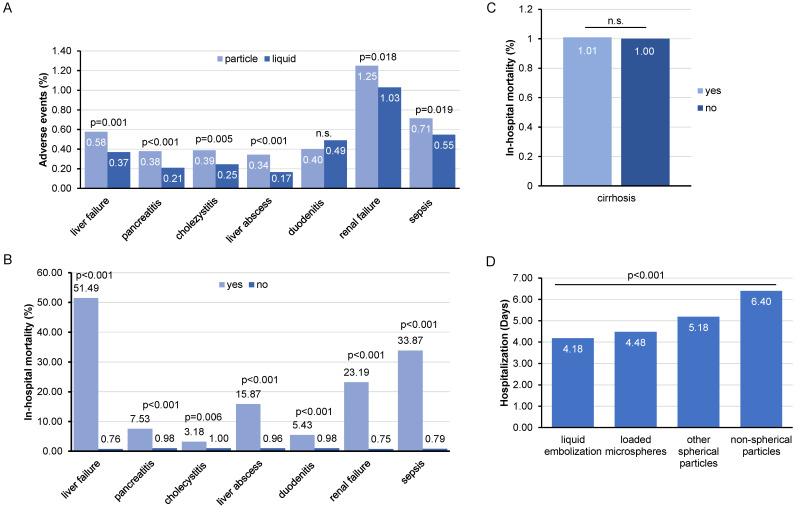
Adverse events and factors associated with an increased in-hospital mortality for TACE. (**A**) Adverse events associated with TACE during hospitalization as a percentage that occurred for embolization with particles versus with liquids. (**B**) Organ complications do increase in-hospital mortality significantly, with the highest increase incidence occurring when liver failure is diagnosed. (**C**) There was no significant difference in in-hospital mortality in patients with or without preexisting liver cirrhosis. (**D**) Total hospital length of stay for TACE considering different embolization agents.

**Figure 4 cancers-14-02088-f004:**
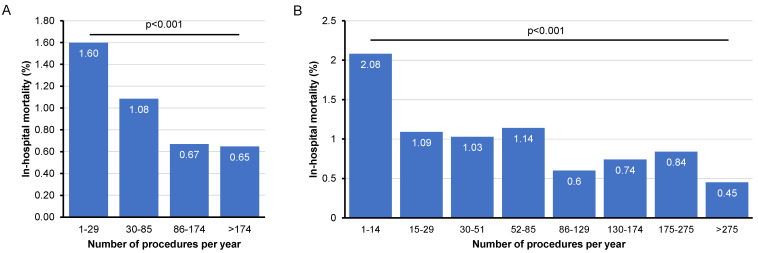
Influence of hospital case volume on in-hospital mortality. (**A**) There is a significant correlation between the number of TACE procedures per year and in-hospital mortality. (**B**) In-hospital mortality decreases above a number of 15 procedures per year.

**Table 1 cancers-14-02088-t001:** Characteristics of study population.

	Study Population
Total number of TACE procedures	49,595
In-hospital death (total)	497
In-hospital mortality rate (%)	1.00
Sex (total)	
Male	36,567
Female	12,028
Age (Mean and SD)	67.71 (10.40)
Age group (total)	
0–17 years	11
18–30 years	112
31–50 years	2853
51–70 years	24,639
>70 years	21,980
Federal state (total)	
Baden-Württemberg	6777
Bavaria	5776
Berlin	3923
Brandenburg	52
Bremen	87
Hamburg	2562
Hesse	5281
Lower Saxony	2930
Mecklenburg-Western Pomerania	497
North Rhine-Westphalia	9162
Rhineland-Palatinate	4258
Saarland	506
Saxony	2660
Saxony-Anhalt	602
Schleswig-Holstein	1352
Thuringia	3170
Underlying diagnosis for TACE (total)	
Hepatocellular Carcinoma (HCC)	33,726
Cholangiocellular Carcinoma (CCA)	13,578
Liver Metastases	2291
Acute or subacute liver failure (total)	
Yes	235
No	49,360
Acute pancreatitis (total)	
Yes	146
No	49,449
Cholecystitis (total)	
Yes	157
No	49,438
Liver abscess (total)	
Yes	126
No	49,469
Duodenitis (total)	
Yes	221
No	49,374
Gastritis (total)	
Yes	1857
No	47,738
Acute kidney failure (total)	
Yes	565
No	49,030
Sepsis (total)	
Yes	313
No	49,282
Cirrhosis (total)	
Yes	16,552
No	49,282
TACE procedures (total)	
Selective embolization with drug-eluting particles	15,670
Selective embolization with non-spherical particles	1160
Selective embolization with spherical particles	7926
Selective embolization with embolizing liquids	24,830
Annual TACE case volume groups based on quartiles (total)	
LVC 1–29 (cases/year)	12,507
MLVC (30–85 cases/year)	12,363
MHVC (86–174 cases/year)	12,369
HVC (>174 cases/year)	12,356
Annual TACE case volume groups based on octiles (total)	
1–14 cases/year	6452
15–29 cases/year	6055
30–51 cases/year	6146
52–85 cases/year	6217
86–129 cases/year	6127
130–174 cases/year	6242
175–275 cases/year	6321
>275 cases/year	6035

## Data Availability

The results of the remote data analyses provided by the Federal Statistical Office are available from the corresponding author on reasonable request.

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
