# Peer review of "Recent Trends and In-Hospital Mortality of Transarterial Chemoembolization (TACE) in Germany: A Systematic Analysis of Hospital Discharge Data between 2010 and 2019"

_cancers, 2022, doi:10.3390/cancers14092088_

Round 1

Reviewer 1 Report

General comments:

The manuscript presents an elegant and comprehensive study of current trends of transarterial chemoembolization (TACE) and its in-hospital mortality in Germany. The aforementioned therapy has become the standard of care and recommended treatment recently due to its survival benefits in subgroups of patients with early to intermediate-stage HCC. The authors identified several factors related to increased in-hospital post-TACE mortality and indicate that their correction might help to reduce future mortality of the procedure. 

Specific comments:

Only small spelling errors were detected.

Conclusion: minor revision is required.

Author Response

We would like to thank this reviewer for her/his positive comments on our manuscript. We have now carefully corrected the remaining spelling errors in the revised version of our manuscript and hope that it will be deemed suitable for publication in its revised form. 

Reviewer 2 Report

Based on retrospective cohorte provided by the German Federal Statistical Office from 2010 to 2019, authors identified a variety of factors such as post-interventional complications, the embolization method used as well as the hospitals’ annual case volume, which areassociated with an increased in-hospital mortality following TACE. It is a very interesting study. Authors showed especially that  in-hospital mortality significantly differed between the underlying indications for TACE (HCC: 0.83%, liver metastases: 1.22%, and CCA: 1.40%) as well as between different embolization agents (liquid embolization: 0.80%, loaded microspheres:0.92%, spherical particles: 1.54% and non-spherical particles: 2.84%), and with number of procedures performed in the center (<15 TACE/year: 2.08% vs. >275 TACE/year: 0.45%). The number of patients included is very impressive and we regret that multivariate analysis have not be used.

The discussion is very interesting does insist very well (with many references) on the question about the efficacy of diferent type of TACE, not resolved by this study.

Author Response

We would like to thank this reviewer for the careful evaluation of our manuscript and the positive feedback on our study. It would certainly be extremely interesting to see which variables emerge as independent prognostic factors following TACE in multivariate regression analysis. However, we must point out that our cohort is composed of cases rather than individual patients. Therefore, a patient in our study may occur more than once as a case during the observed study period. This could significantly bias the results of a multivariate analysis. Furthermore, the primary aim of our study was to investigate current trends of TACE and its in-hospital mortality in Germany. Nevertheless, it will certainly be an extremely important issue in the future to provide evidence for the superiority of a specific TACE technique and to find out which factors, among others, influence hospital length of stay and morbidity related to TACE. To fully comply with this reviewer’s comment, we have now added a respective statement on the missing multivariate analysis within the revised discussion section of our manuscript as follows:

„These future studies should also include multivariate analyses to further dissect the influence of individual factors on in-hospital mortality.“

Round 2

Reviewer 2 Report

 Does the authors response: "However, we must point out that our cohort is composed of cases rather than individual patients" mean that one patient may have undergone different type of embolization's agents? or have been treated in different center (with different number of procedures per year)? If yes, proposed univariate analysis are also questionable!

Author Response

We would like to thank this expert for his/her inquiry regarding our data source. The data we used as a basis for our study are completely anonymized and are provided by the Federal Statistical Office for remote data analysis. These data consist of individual hospital cases, which are automatically transmitted from the hospital to the Federal Statistical Office upon discharge. This means that we cannot identify an individual patient, but can only look at the anonymized cases. Based on this fact, it is quite possible that a HCC patient initially received TACE in a first hospital (case 1) and after e.g. 4 months received the second TACE in another hospital with a different agent (case 2). Therefore, our analyses can only refer to the case level and analyze the mortality of each individual hospitalization. To fully comply with this reviewer's comment, we have addressed this imporant issue in the revised limitation section of our manuscript as follows:

"Finally, our data are based on individual cases/TACE procedures and not individual patients. Therefore, no information was available on the number of patients who underwent TACE more than once"